# B cell maturation antigen (BCMA) is dispensable for the survival of long-lived plasma cells

Shannon R. Menzel[1], Edith Roth[1], Jens Wittner[1], Stefanie Brey[2], Leonie Weckwerth[1], Jana Thomas[1], Thomas H. Winkler ®[2], Wolfgang Schuh[1], Hans-Martin Jäck ®[1] ✉, Katharina Pracht ®[1,3] & Sebastian R. Schulz[1,3]

The survival of antibody-secreting plasma cells is essential for long-lasting humoral immunity. BCMA is proposed to promote APRIL-mediated survival signals. However, extensive shedding of murine BCMA raises doubts about its role as a signaling receptor. To unequivocally establish BCMA's function in plasma cell survival, we generate two BCMA-deficient mouse lines and examine antigen-specific plasma cells post-immunization. Contrary to previous reports, both BCMA-deficient mouse lines have comparable numbers of antigen-specific long-lived plasma cells following both protein and mRNA immunizations. Transcriptome analysis reveals no reduction in survival signaling upon BCMA deletion. Interestingly, BCMA-deficient mice show increased total plasma cell numbers in the bone marrow and mesenteric lymph nodes after boost immunizations. These results indicate that BCMA has no intrinsic role in maintaining long-lived plasma cells. Instead, we propose that BCMA's function is limited to acting as a soluble decoy receptor for APRIL, thereby fine-tuning the plasma cell population size by limiting survival factor availability. Our findings thus provide a strong argument against the APRIL-BCMA axis being a central mechanism for plasma cell longevity.

Antibody-secreting plasma cells (ASC) confer protection after vaccination but also contribute to pathogenicity in autoimmunity and plasma cell dyscrasias. Therefore, plasma cells are intriguing targets in both health and disease. The lifespan of an individual mouse and human plasma cell can range from days to years, as shown by kinetic measurements of serum antibody concentrations, genetic time-stamping approaches, and radiocarbon dating of plasma cell populations[1–4].

Current models of sustained ASC survival suggest that plasma cells settle into sanctuary sites or dedicated niches, where cell-cell contacts, soluble factors, and intrinsic regulators work in concert to promote plasma cell persistence. While the fate decisions guiding plasma cells into these niches are still incompletely understood, the tumor necrosis factor (TNF) family member A proliferation-inducing ligand [APRIL: encoded by tumor necrosis factor ligand superfamily member (*Tnfsf*) *13*] has emerged as a critical extrinsic factor in plasma cell maturation and longevity. APRIL and the related family member B-cell activating factor (BAFF, encoded by *Tnfsf13b*) form an intricate interaction network with their receptors BAFF-Receptor [BAFFR, encoded by tumor necrosis factor receptor superfamily member (*Tnfrsf*) *13c*], Transmembrane activator and CAML interactor (TACI, encoded by *Tnfrsf13b*), and B cell maturation antigen (BCMA, encoded

[1]Division of Molecular Immunology, Department of Medicine 3—Rheumatology and Immunology, Friedrich-Alexander-University Erlangen-Nürnberg, Erlangen, Germany. [2]Division of Genetics, Department Biology, Nikolaus-Fiebiger-Center for Molecular Medicine, Friedrich-Alexander-University Erlangen-Nürnberg, Erlangen, Germany. [3]These authors contributed equally: Katharina Pracht, Sebastian R. Schulz. ✉e-mail: hans-martin.jaeck@fau.de

by *Tnfrsf17*[5,6], additionally relying on sequestration by heparan sulfate proteoglycans like CD138 (Syndecan 1, *Sdc1*) for its survival-promoting effects on plasma cells[7].

BCMA displays the highest affinity for APRIL among the TNF-receptor family members[8,9], rendering this receptor a prime candidate for mediating survival signaling in plasma cells. In a recent reporter mouse model, we demonstrated a highly restricted expression of BCMA-encoding *Tnfrsf17* in CD138+TACI+ ASCs, underlining a proposed role in the final differentiation steps required to enter the long-lived plasma cell compartment[10]. Previous work examining functional consequences of BCMA deficiency in mice has produced conflicting results regarding plasma cell numbers in homeostasis[11–13]. In addition, only sparse data from a single study indicate a reduction of antigen-specific plasma cells in BCMA-deficient mice after primary immunization[14].

To determine the functional role of BCMA in long-term humoral immunity, we generate two independent BCMA-deficient mouse lines and monitor plasma cell persistence and durability of serum antibody concentrations following multiple immunization regimens. In contrast to previously published reports, our analyses did not reveal significant effects of BCMA-deficiency on long-term plasma cell survival in any model. BCMA is, therefore, dispensable for the survival of long-lived plasma cells.

## Results

### BCMA is dispensable for plasma cell homeostasis

To determine whether BCMA controls plasma cell survival, we established a BCMA-deficient mouse model by crossing the recently established BCMA:Tom reporter mouse line[10] to E2A-Cre mice[15], resulting in a germline deletion of *Tnfrsf17* exon 3. This mutant line is referred to as BCMA-KOΔ3 (Fig. 1A). The third exon of the *Tnfrsf17* gene encodes BCMA's intracellular domain with TNF receptor-associated factor (TRAF)-binding motifs required for signaling via nuclear factor 'kappa-light-chain-enhancer' of activated B-cells (NF-κB)[16]. The BCMA knock-out was confirmed using flow cytometry, evidenced by the lack of surface BCMA on splenic CD138+TACI+ antibody-secreting cells (ASCs) in BCMA-KOΔ3 mice. These spleen suspensions were treated with the γ-secretase inhibitor DAPT (*N*-[*N*-(3,5-Difluorphenacetyl)-L-alanyl]-*S*-phenylglycin-tert-butylester), which blocks BCMA shedding and allows its detection by flow cytometry (Fig. 1B and Supplementary Fig. S1A). Additionally, transcriptomic analysis of ASCs from the bone marrow of BCMA-KOΔ3 mice revealed no read counts for the deleted *Tnfrsf17* exon 3 and strongly reduced read counts of the non-deleted exons (Supplementary Fig. S1B), indicating the complete loss of BCMA in this mouse model. The deletion of *Tnfrsf17* exon 3 affected neither serum immunoglobulin (Ig) M and IgG concentrations in non-immunized mice (Supplementary Fig. 1C) nor frequencies of the total CD138+TACI+ ASC population in bone marrow and spleen, the maturation-associated ASC subsets P0-P3[17], and the Ig heavy (H) chain isotype distribution in ASCs (Fig. 1C and Supplementary Fig. S1D). Serum IgA concentrations and total ASC numbers in the mesenteric lymph nodes were slightly increased (Supplementary Fig. S1C, D). Overall, these findings align with the initial characterization of a previously published BCMA-deficient mouse line with a similar germline deletion of exon 3, which also showed no changes in splenic CD138+ B cells[13]. However, while subsequent studies with the same non-immunized BCMA-deficient mouse line reported reduced frequencies of CD138+ bone marrow plasma cells[12], we could not replicate any changes in the non-immunized ASC compartment in our BCMA-KOΔ3 mice.

BCMA could not be detected on the surface of murine plasma cells by flow cytometry due to cleavage and shedding mediated by γ-secretase[18] (Fig. 1B and Supplementary Fig. S1A). The absence of the extracellular APRIL-binding domain of BCMA on the plasma cell surface challenges the proposed role of BCMA as a signal-transducing surface receptor with a pro-survival function for maintaining long-lived plasma cells. To address this question, we compared the transcriptome of bone marrow and splenic ASCs from wild-type and BCMA-KOΔ3 mice (Fig. 1D). Principal component analysis revealed no clustering of BCMA-deficient and WT ASCs in the bone marrow. Similarly, there was no clear separation of the splenic ASC transcriptomes according to genotype. These data indicate that BCMA-deficient and WT ASCs share almost identical transcriptional profiles. In support, no differentially expressed genes other than *Tnfrsf17* (encoding BCMA) were detected between BCMA-KOΔ3 and WT ASCs in both tissues. We were also unable to confirm the published finding that the expression of the anti-apoptotic myeloid cell leukemia sequence *(Mcl)−1* is reduced in mature BCMA-deficient bone marrow plasma cells[12]. Based on our findings that BCMA deficiency neither affected transcriptome profiles of ASCs nor plasma cell numbers and total serum antibody concentrations, we conclude that BCMA is dispensable for the development, maintenance, and function of ASCs in non-immunized mice.

### BCMA is dispensable for sustained humoral immune responses

Reduced numbers of antigen-specific ASCs were detected in a BCMA-deficient mouse line 7 weeks after T-dependent immunization with a hapten-carrier antigen[14], implicating BCMA in APRIL-mediated survival of long-lived bone marrow plasma cells. We attempted to reproduce this key experiment, which, to our knowledge, stands as the sole piece of evidence supporting BCMA's role as a survival receptor for long-lived plasma cells. Therefore, we immunized BCMA-KOΔ3 and control mice with the hapten-carrier conjugate NP-KLH (4-Hydroxy-3-nitrophenyla-cetyl-Keyhole Limpet Hemocyanin) in alum and quantified both antigen-specific and total ASC numbers 7 weeks after immunization using an enzyme-linked immunoassay (ELISA) Spot assay and flow cytometry (Fig. 2A). The kinetics of NP-specific serum IgG concentrations were comparable between BCMA-KOΔ3 and wildtype mice up to day 49 (Fig. 2B), which aligns with previous reports[13,14]. However, in contrast to the study by O'Connor et al., we detected comparable numbers of NP-specific ASCs in the bone marrow and spleen after 49 days, irrespective of their IgH chain isotype (Fig. 2C and Supplementary Fig. S2A). Further, the number of total ASCs was unaltered in all analyzed tissues of BCMA-KOΔ3 compared to wildtype mice (Fig. 2D).

Recent data suggest that long-lived plasma cells are generated continuously during a germinal center (GC) response[3] and that GC responses can be maintained for several weeks to months after antigen encounter[19]. Therefore, a reduction of NP-specific ASCs in BCMA-KOΔ3 mice could conceptually be compensated by a prolonged output of ASCs from the GC. Yet, by flow cytometric analysis, we detected only very few antigen (NP)-specific GC B cells remaining 7 weeks after the initial immunization, with no significant differences in the frequencies of total GC B cells between BCMA-KOΔ3 and wildtype mice (Fig. 2E). Therefore, BCMA-deficiency does not prolong GC persistence and does not increase the output of antigen-specific ASCs. Instead, the comparable numbers of long-lived ASCs in BCMA-deficient mice are likely due to similar survival rates of the plasma cells generated during the primary immune response rather than enhanced turnover or replenishment. This is further supported by comparable frequencies of Ki67+ proliferating ASCs in BCMA-KOΔ3 and WT mice (Supplementary Fig. S2B). As BCMA, TACI also binds to APRIL and BAFF and could serve as the primary or compensatory survival-mediating receptor in the absence of BCMA. We did not detect increased *Tnfrsf13b* (encoding TACI) expression by transcriptomic analysis in ASC from non-immunized mice (Fig. 1E). In addition, TACI surface abundance was similar on ASCs from WT and BCMA-deficient mice after primary immunization, further indicating TACI's survival capacity without the necessity to upregulate TACI as a compensatory APRIL receptor (Supplementary Fig. S2C).

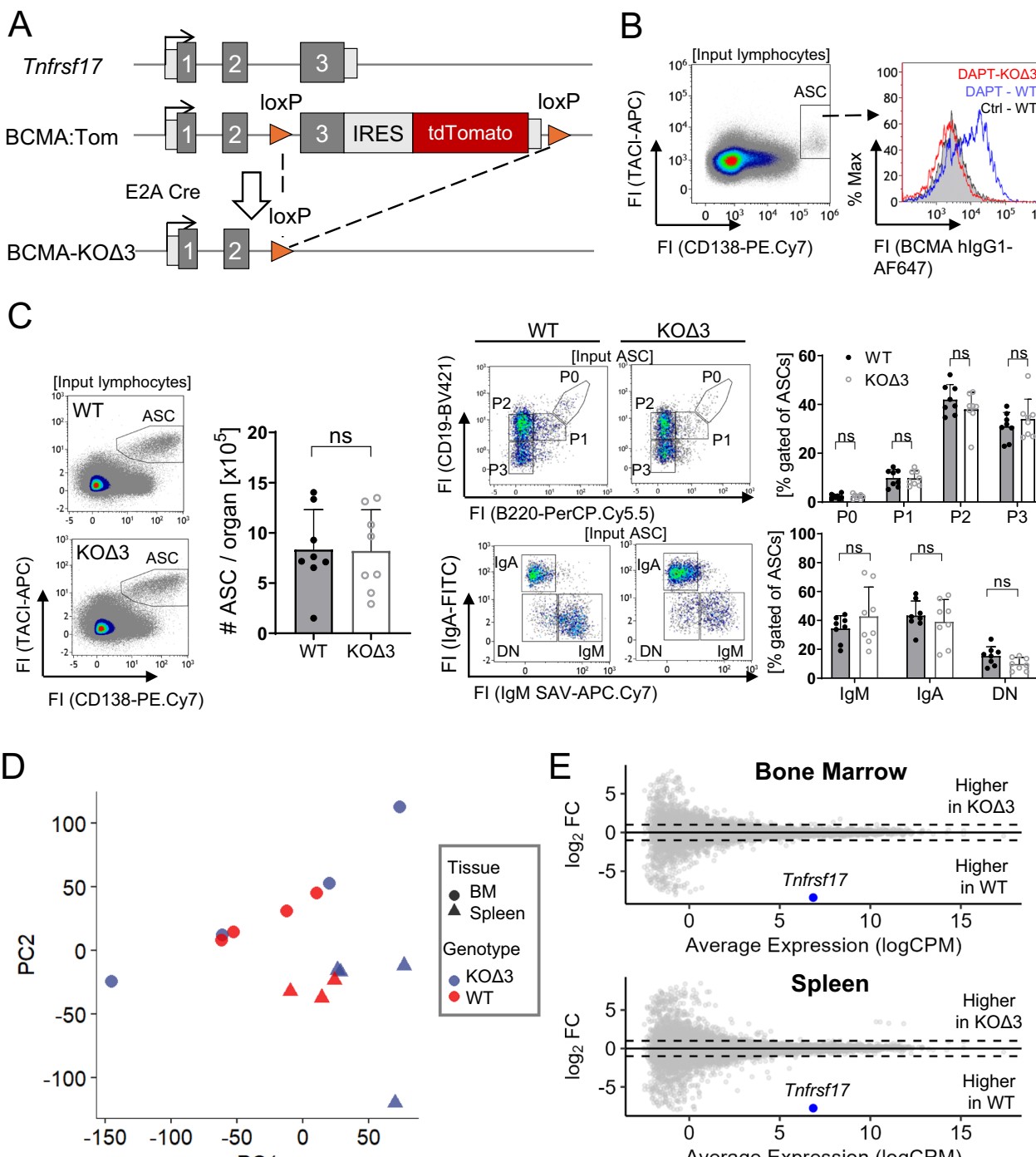

**Fig. 1 | Characterization of antibody-secreting cell subsets in BCMA-KOΔ3 mice.**
**A** Schematic illustration of the genetic locus of BCMA-KOΔ3 mice. BCMA:Tom mice with loxP sites flanking exon 3 of the BCMA-encoding *Tnfrsf17* gene and the IRES-tdTomato cassette were crossed to E2A-Cre mice, resulting in a germline deletion of exon 3. Exons are illustrated in gray boxes, and light gray boxes indicate 5' and 3' UTRs. **B** Flow cytometric analysis of BCMA cell surface abundance on splenic ASCs after γ-secretase inhibitor (DAPT) treatment using the anti-BCMA antibody clone 25C2. Splenic single-cell suspensions were cultured for 18 h with 1 μM DAPT or DMSO solvent control (Ctrl). **C** Representative gating strategy to quantify frequencies of CD138$^+$TACI$^+$ ASCs, ASC subsets P0 (B220$^{hi}$CD19$^{hi}$), P1 (B220$^+$CD19$^+$), P2 (B220$^-$CD19$^+$), and P3 (B220$^-$CD19$^-$) and IgH-chain isotype distribution in the bone marrow (BM) ASCs. Double negative (DN) ASCs contain mostly IgG ASCs[10,17]. *n* = 8

mice per group from 3 independent experiments. Bar diagrams show mean and SD with each dot indicating one mouse. **D, E** Transcriptome analysis of bone marrow and splenic ASCs isolated from non-immunized BCMA-KOΔ3 and wildtype mice (WT). Principal component analysis (**D**) visualizes sample similarities; differential gene expression is visualized in the MA plot (**E**). In BCMA-KOΔ3 ASCs, no upregulated genes were detected, and the only downregulated gene (*Tnfrsf17*) is colored in blue. Statistical analysis in (**C**) was performed with a two-tailed unpaired *t*-test to compare total ASC numbers. Comparisons of ASC subsets were conducted by two-way ANOVA with Šídák's correction for multiple comparisons. Exact *p*-values, mouse sex, and ages are provided in the Source Data file. ns not significant, ASC antibody-secreting cell, PC principal component.

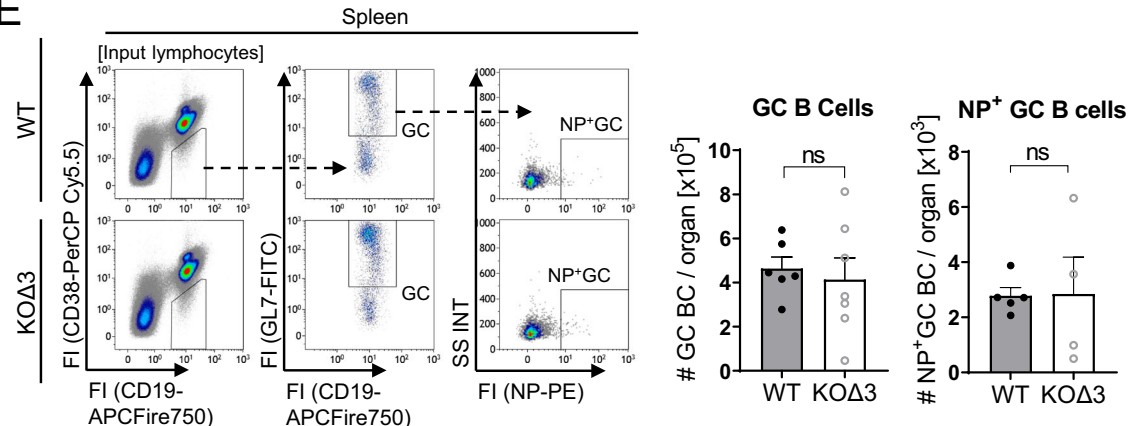

Despite following a similar experimental procedure, we could not reproduce the foundational observation that established BCMA as a survival factor for long-lived plasma cells[14]. To minimize the influence of artifacts introduced into the *Tnfrsf17* locus by the BCMA-KOΔ3 conditional deletion and to completely exclude the residual expression of a truncated BCMA, we generated a second, independent BCMA-deficient mouse model with genomic deletion of the entire *Tnfrsf17*

locus (BCMA-KO) (Supplementary Fig. S3A). Serum IgM, IgA, and IgG concentrations were comparable between non-immunized BCMA-KO and WT mice (Supplementary Fig. S3B). These BCMA-KO mice were immunized with NP-KLH according to the same regimen as BCMA-KOΔ3 mice and analyzed after seven weeks (Fig. 2A). Confirming our results derived from the BCMA-KOΔ3 mouse model, NP-specific serum IgG concentrations and NP-specific bone marrow ASC numbers of all

**Fig. 2 | Quantification of antigen-specific and total ASCs in BCMA-KOΔ3 and WT mice after T-dependent immunization with NP-KLH. A** Schematic illustration of the experimental setup. BCMA-KOΔ3 and WT control mice were immunized with 100 μg NP-KLH in alum and analyzed 7 weeks after immunization. **B** NP-specific IgG serum concentration was determined by ELISA for WT (black, $n = 9$) and BCMA-KOΔ3 mice (gray, $n = 6$). **C** IgH-chain isotype-specific quantification of antigen-specific ASCs by ELISpot analysis in bone marrow. The images below are representative pictures of ELISpot analysis with numbers indicating the number of cells seeded per well (NP-IgM: $n = 7$ (WT), 5 (KOΔ3); NP-IgA: $n = 9$ (WT), 6 (KOΔ3); NP-IgG: $n = 8$ (WT), 6 (KOΔ3)). **D** Flow cytometric quantification of total ASC numbers per organ (Bone marrow was pooled from one femur and tibia/mouse, $n = 9$ WT and 10 KOΔ3 for bone marrow and splenic ASCs; $n = 8$ WT and 10 KOΔ3 for mLN ASCs). **E** Flow cytometric quantification of germinal center (GC) B cells and antigen-specific GC B cells (NP⁺ GC B cells) numbers in the spleen. Bar diagrams show mean and SD with each dot indicating one mouse (GC B cell: $n = 6$ WT mice and $n = 7$ BCMA-KOΔ3 mice, NP⁺ GC B cell: $n = 5$ WT mice and $n = 4$ BCMA-KOΔ3 mice). All statistical comparisons were performed using a two-way ANOVA with Šídák's multiple comparisons test. Exact $p$-values, mouse sex, and ages are provided in the Source Data file. ns not significant, ASC antibody-secreting cell, i.p. intra-peritoneal, mLN mesenteric lymph node, BC B cell.

IgH-chain isotypes were again indistinguishable in BCMA-KO mice and wildtype controls at all analyzed time points (Supplementary Fig. S3C, D). Further, we assessed the impact of BCMA deficiency on the maintenance of high-affinity plasma cells. However, the binding of serum IgG to low- and high-valency antigen (NP) determined by ELISA at day 49 after immunization was comparable between BCMA-KO and wildtype mice (Supplementary Fig. S3E), suggesting that BCMA-deficiency does not impact the survival capacity of high-affinity plasma cells. Therefore, we conclude that BCMA is dispensable for the longevity of plasma cells generated in a primary immune response.

### BCMA is dispensable for plasma cell longevity
In contrast to the study by O'Connor and colleagues[14], we could not detect a reduction of antigen-specific ASCs in BCMA-deficient mice 7 weeks after primary immunizations (Fig. 2C and Supplementary Fig. S3D). To additionally analyze the effect of BCMA deficiency on the survival of plasma cells that were generated during a memory response, we boosted BCMA-KOΔ3 and wildtype mice at day 42 after the primary immunization with NP-KLH and euthanized the animals at day 128 (Fig. 3A). The increase of antigen-specific IgG serum concentrations confirmed the initiation of a memory immune response after the boost with comparable peak concentrations and decay curves between BCMA-KOΔ3 and wildtype mice throughout the experiment (Fig. 3B). Accordingly, similar numbers of NP-specific ASCs were detected in the bone marrow for all IgH-chain isotypes (Fig. 3C). Furthermore, IgG antigen affinity assessed by the ratio of $NP_{(4)}/NP_{(20)}$-binding again indicated no impact of BCMA-deficiency on high-affinity plasma cell survival (Supplementary Fig. S3F). This again supports our conclusion that BCMA is dispensable for maintaining long-lived plasma cells.

Surprisingly, the total number of ASCs significantly increased 2-fold in the bone marrow and mesenteric lymph nodes but not in the spleens of boosted BCMA-KOΔ3 mice (Fig. 3D). This increase in total ASC numbers was driven by an expansion of IgA⁺ and IgM⁺ ASCs, while the number of IgG⁺ ASCs did not significantly increase (Fig. 3E, F). The observed increase in total ASCs without an accompanying rise in antigen-specific ASCs suggests an expansion of non-antigen-specific or bystander ASCs after the boost immunization. This expansion could be driven by the increased availability of the survival factor APRIL. Soluble BCMA shed from the surface of ASCs acts as a decoy receptor for APRIL[18], thereby potentially limiting APRIL availability and subsequently TACI-mediated survival signaling in ASCs. In the absence of soluble BCMA, the increased availability of APRIL (and/or BAFF) at the sites of plasma cell induction or within the survival niches could enhance the persistence of bystander or non-antigen-specific ASCs generated in the absence of specific antigen stimulation.

To verify that BCMA is dispensable for plasma cell longevity in memory responses, we repeated the prime-boost regimen by immunizing BCMA-KOΔ3 and wildtype mice with the severe acute respiratory syndrome coronavirus 2 (SARS-CoV-2) vaccine mRNA-1273 (Fig. 4A). Detection of SARS-CoV-2 receptor-binding-domain (RBD)-specific IgG serum concentrations confirmed the induction of an immune response and a substantial boost reaction that was again comparable in amplitude and kinetics between BCMA-KOΔ3 and wildtype mice (Fig. 4B). Accordingly, numbers of RBD-specific ASCs quantified by flow cytometric tetramer staining were equivalent in BCMA-KOΔ3 and wildtype bone marrow and spleens (Fig. 4C and Supplementary Fig. S4A). These findings confirm our results from NP-KLH immunization experiments (Fig. 3C) and support our conclusion that BCMA has no intrinsic role in maintaining long-lived plasma cells.

We again detected a ~60% increase in the total ASC population in the bone marrow from mRNA-1273 boost-immunized BCMA-KOΔ3 mice compared to WT animals (Fig. 4D). The increase was again predominantly driven by non-RBD-specific ASCs, with IgA⁺ ASCs contributing most significantly and IgM⁺ ASCs to a lesser extent (Fig. 4E and Supplementary Fig. S4B, C). Serum antibody concentrations mirrored the increase of total IgA⁺ and IgM⁺ ASCs (Fig. 4F). In contrast, IgA detected in feces was unchanged (Fig. 4F). To determine whether alterations of intrinsic BCMA-dependent signaling cascades mediate the global increase in BCMA-deficient ASCs, we performed transcriptome profiling of bone marrow ASCs isolated from mRNA-1273-immunized BCMA-KOΔ3 and WT mice. Again, the major principal components failed to capture genotype-related differences between the samples from BCMA-KOΔ3 and wildtype mice, suggesting only a limited impact of BCMA-deficiency on the ASC transcriptome after boost immunization (Fig. 4G). Once more, we detected *Tnfrsf17* as the only gene with differential expression between wildtype and BCMA-KOΔ3 cells (Fig. 4H). Therefore, the gene expression profiles of the plasma cells that populate the bone marrow of BCMA-deficient mice in increased numbers are unaffected by the loss of BCMA and mirror those of their wild-type counterparts.

## Discussion
Despite widespread reports that BCMA controls plasma cell survival, only one study supports BCMA's intrinsic role in the survival of long-lived plasma cells after immunization[14], with other studies reporting inconsistencies in its function in plasma cell biology under steady-state conditions[11–13,20]. Some of these studies used BCMA-deficient mice of mixed genetic backgrounds or from commercial vendors without information on the genomic locus details[13,14]. In this study, we aimed to resolve these discrepancies by replicating the experiment that proposed the fundamental role of BCMA for long-lived plasma cell survival in two independent, well-characterized BCMA-deficient mouse models. Unexpectedly, we could not reproduce the key data underlying the proposed function of BCMA in mediating the APRIL-dependent survival of long-lived plasma cells. Using multiple immunization regimens, we find not only unaffected initiation of a humoral immune response as described before[13], but also comparable survival of antigen-specific ASCs after primary and memory responses in BCMA-deficient mice. An increased turnover of BCMA-deficient plasma cells as the cause of unaltered antigen-specific ASCs is unlikely, as ASCs presented without detectable transcriptome perturbations in the absence of BCMA. Therefore, membrane-anchored BCMA does not contribute to signaling networks that control the longevity of plasma cells and antibody responses. While previous work found reduced numbers of bone-marrow plasma cells already under steady-state conditions[12,20], we observed largely unaltered ASC compartments and serum antibody concentrations in our BCMA-deficient mice. Bone marrow and

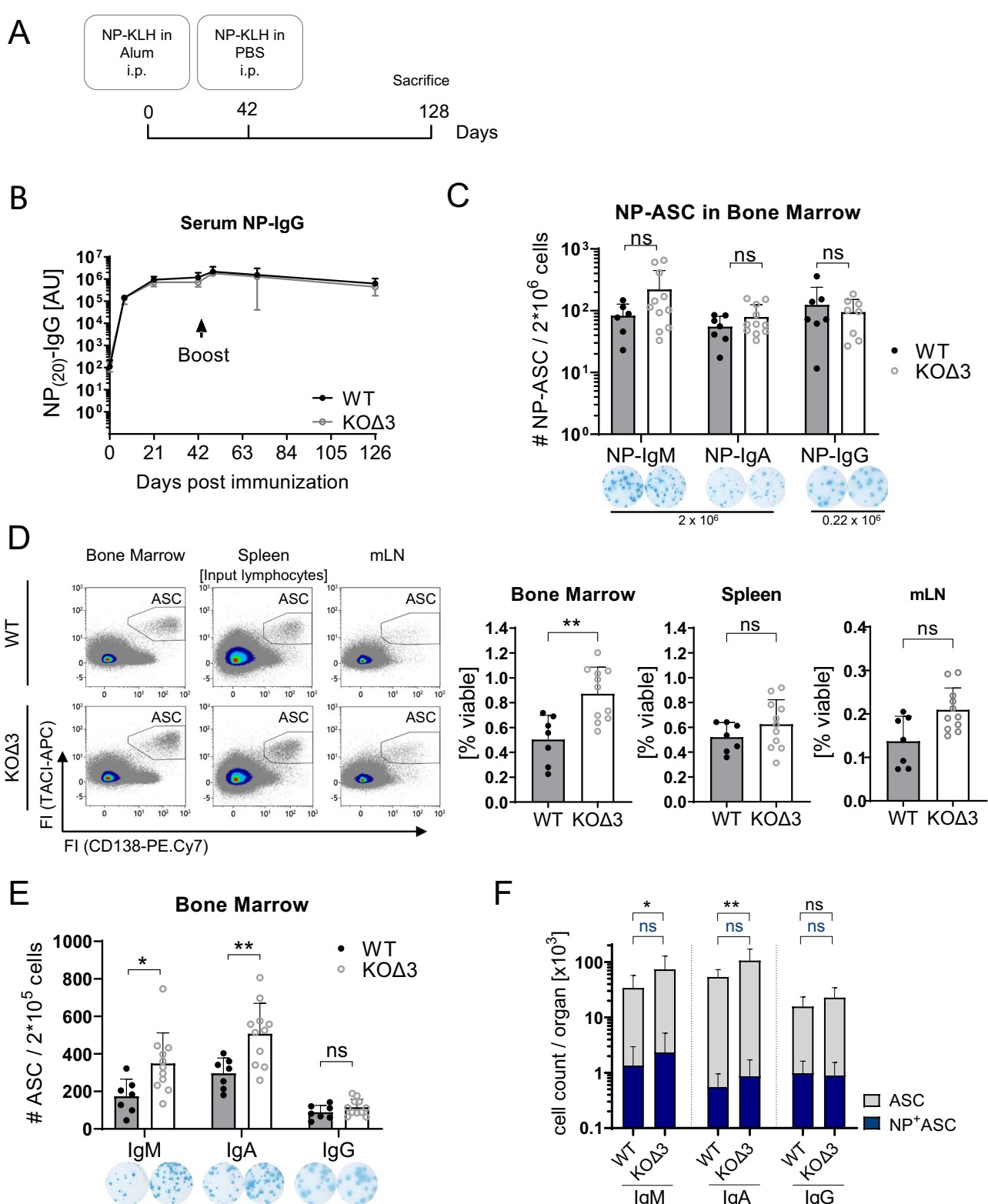

spleen ASC numbers were unaffected in both BCMA-deficient mouse models, while BCMA-KOΔ3 mice had slightly increased numbers of ASC in the mesenteric lymph nodes, accompanied by elevated serum IgA abundance. The underlying mechanisms remain unclear but appear not to be a consequence of the loss of BCMA, as our second BCMA-deficient mouse line does not display these alterations, similar to the observations of the established BCMA-KO mice by Eslami and colleagues[11].

APRIL is a crucial factor promoting plasma cell survival in vitro[21,22] and seems to share redundant roles with BAFF in vivo[11,23]. Among APRIL's receptors, BCMA has been the predominant candidate for mediating this effect on plasma cells. This conclusion is based on the previous finding of reduced antigen-specific ASC numbers in immunized BCMA-deficient mice in combination with the assumption of low *Tnfrsf13b* expression (encoding TACI) in ASCs[14] and the higher binding affinity of APRIL to BCMA compared to TACI[5,24]. However, our data

**Fig. 3 | Quantification of antigen-specific and total ASCs in BCMA-KOΔ3 and WT mice after secondary T-dependent immunization with NP-KLH. A** Schematic illustration of the experimental setup. BCMA-KOΔ3 and control mice were immunized with 100 µg NP-KLH in alum, boosted with 50 µg NP-KLH in PBS on day 42, and analyzed on day 128. **B** NP-specific IgG serum concentrations were determined by ELISA for wildtype (WT) (black, $n = 7$) and BCMA-KOΔ3 mice (gray, $n = 8$). **C** IgH-chain isotype-specific quantification of antigen (NP)-specific ASCs by ELISpot analysis in bone marrow. The images below are representative pictures of ELISpot analysis with numbers indicating the number of cells seeded per well ($n = 6$ WT and 11 KOΔ3 for NP-IgM; $n = 7$ WT and 11 KOΔ3 for NP-IgA; $n = 7$ WT and 8 KOΔ3 for NP-IgG). **D** Flow cytometric quantification of total ASC numbers per organ (bone marrow was pooled from one femur and one tibia/mouse, $n = 7$ WT and 11 KOΔ3).

**E** IgH-chain isotype-specific quantification of total bone marrow ASCs by ELISpot analysis. The numbers below the ELISpot images indicate the number of seeded cells per well ($n = 7$ WT and 11 KOΔ3). **F** Stacked bar diagram with combined data from C and E. Bar diagrams show mean and SD with each dot indicating one mouse. Statistical analysis in B and F was performed using a two-way ANOVA with Šídák's multiple comparisons test. Statistical analysis (**C–E**) was performed with unpaired $t$-tests, correcting for multiple comparisons by the false discovery rate (FDR) according to Benjamini, Krieger, and Yekutieli's Two-stage step-up Method. Exact $p$-values, mouse sex, and ages are provided in the Source Data file. ASC antibody-secreting cell, WT wildtype, ns not significant, i.p. Intra-peritoneal, mLN mesenteric lymph node, $^*p \le 0.05$, $^{**}p \le 0.01$.

clearly show that the APRIL-BCMA axis is dispensable for long-lasting plasma cell survival, highlighting alternative, BCMA-independent mechanisms that enable the long-term persistence of plasma cells, e.g., through signaling via TACI. TACI binds both cytokines, BAFF and APRIL[8], which, together with our data, suggests that TACI may play a more prominent role in supporting plasma cell survival than BCMA. However, interpreting the role of TACI specifically in plasma cell biology is limited as only mouse models carrying genomic deletions of TACI have been used to date[11,25,26]. As TACI is already expressed in antigen-activated B cells[27], these models exhibited a phenotype of increased lymphoproliferation and autoimmunity[26,28] and altered ASC differentiation[25,29], necessitating cautious interpretation of TACI's role in plasma cell generation, maintenance, and long-term survival. Thus, further investigations, e.g., by using targeted strategies that enable a conditional deletion of TACI exclusively in plasma cells, are needed to elucidate the precise role of TACI in plasma cell biology and its interplay with APRIL.

While the genomic deletions of BCMA did not result in differential survival of long-lived ASCs after both primary and boost immunizations with NP-KLH or mRNA-1273, total bone marrow ASC numbers unexpectedly doubled in BCMA-deficient compared to WT mice after boost immunizations. This significant increase was primarily driven by non-antigen-specific or bystander ASCs, with a particular increase in IgM+ and IgA+ ASCs. The bulk transcriptional profiles of these ASCs were indistinguishable from their WT counterparts, suggesting that the observed expansion is not due to intrinsic mechanisms, e.g., increased survival signaling or enhanced proliferation, but may instead be driven by extrinsic factors resulting in a more permissive environment for ASC survival. Given that the cleaved extracellular fragment of BCMA binds and masks the pro-survival factor APRIL[18], this APRIL-decoy function may be the primary role of murine BCMA. Without this soluble decoy, increased availability of APRIL in the ASC microenvironments may increase the survival probabilities and longevity of ASCs in the bone marrow[30]. This proposed mechanism might also control the lymphoproliferation and significantly increased numbers of plasma cells observed in lupus-prone mice with a BCMA-deficiency[20,31]. However, these changes were primarily detected in the spleen and lymph nodes. The reason why the ASC expansion is detectable only after booster immunization and not during the primary immune response in our BCMA-deficient mice remains unclear. Repeated exposure to the antigen in a booster immunization could amplify any subtle effects of BCMA-deficiency that were not apparent during the primary response.

The concept that BCMA is critical for human plasma cell survival relies largely on the extrapolation of murine data[14] to the human system[32]. In contrast to the well-established role of APRIL in driving human ASC maturation[33], direct experimental evidence of BCMA's role in human ASCs remains limited. The available data primarily relies on multiple myeloma (MM) cell lines as the malignant counterparts of normal plasma cells. While initial studies reported decreased survival of MM cells upon deletion or knockdown of BCMA[34], more recent findings have challenged this assumption, showing no significant

impact of BCMA loss on MM cell viability in vitro[35]. Additionally, key species-specific differences in *Tnfrsf17* expression between mice and humans complicate direct translation, including differences in its expression, even outside the ASC compartment[36], and surface retention or cleavage of BCMA.

In contrast to murine BCMA, human BCMA contains a glycosylation site[37] that influences its stability and shedding dynamics[38]. This results in increased surface retention of human BCMA[39] compared to murine BCMA[10], allowing the successful targeting of BCMA-positive cells in human autoimmune and malignant diseases by BCMA-specific antibodies and chimeric antigen receptor (CAR) T cells[40,41]. In light of our data, previous assumptions about the role of BCMA in human plasma cell biology will need to be reassessed.

In summary, by employing two independent BCMA knock-out mouse models and two different immunization regimens, we provide convincing evidence that BCMA is not required for long-lived plasma cell survival in mice. These findings eliminate the APRIL-BCMA axis as a central mechanism for plasma cell longevity.

## Methods
### Mice
C57BL/6N mice were purchased from Janvier (Le Genest Saint Isle, France, stock no. C57BL/6NRj). BCMA-KOΔ3 mice were established by crossing the BCMA:Tom reporter mouse[10] with a Transcription Factor E2-Alpha (E2A)-cre deleter line[15]. After successful recombination, a line without the E2A-Cre transgene was established for further analysis.

All mice were maintained under specific-pathogen-free conditions in the Preclinical Experimental Animal Center (PETZ) or the Nikolaus-Fiebiger Center animal facility of the University of Erlangen-Nürnberg. Mice were housed at temperatures between 22 °C and 23 °C, a humidity of 50–60% and a regulated 12-h light/dark cycle with free access to a chow diet and water. We used age and sex-matched mice of both sexes for all analyses. In all mouse experiments, control and experimental animals were co-housed and euthanized by carbon dioxide inhalation. All animal experiments were performed according to institutional and national guidelines and were approved by the Amt für Veterinärwesen und gesundheitlichen Verbraucherschutz der Stadt Erlangen, Regierung von Unterfranken, Würzburg, Germany.

### CRISPR-Cas-mediated construction of BCMA-deficient (BCMA-KO) mice
To generate BCMA-KO mice, the *Tnfrsf17* locus on mouse chromosome 16 was targeted using two guide RNAs (crRNAs; binding sequence 5′ GUUUGCUGUGAUAUACCCCU 3′, 5′ACCUUGAUCGACAGAUCUGG 3′). The crRNAs, annealed to tracrRNA, and Cas9 protein (all obtained from Integrated DNA Technologies) were injected into the pro-nuclei of fertilized one-cell stage embryos isolated from C57BL/6J breeders. These embryos were then transferred into pseudo-pregnant recipient mice. Viable pups born from the recipient mice were screened for gene deletion by PCR. Targeted animals were backcrossed twice to WT C57BL/6J mice to eliminate off-target mutations. Primers were used for genotyping the BCMA wildtype (5′-ATAAATGGCTACTGCACTTTCG

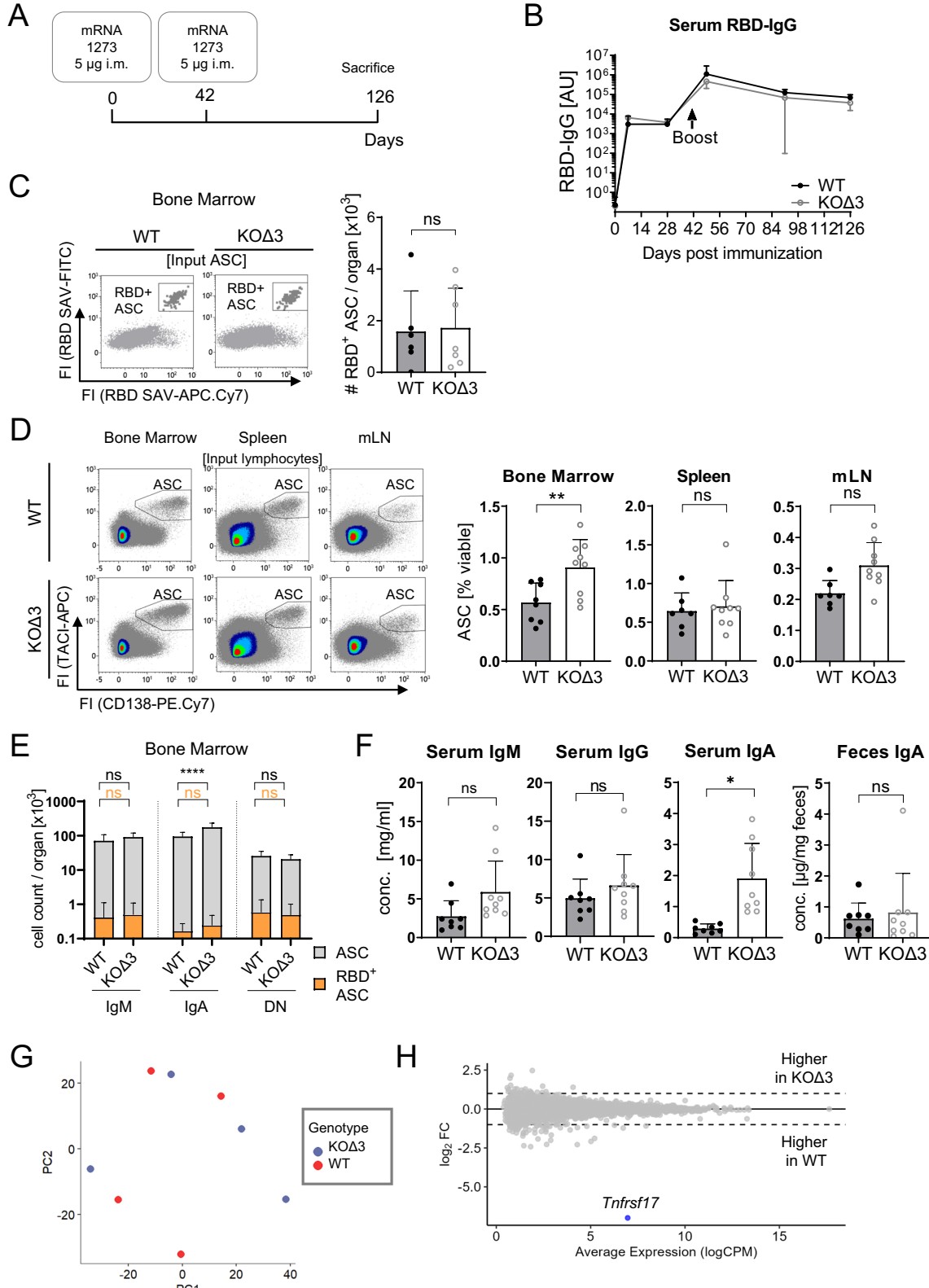

GC-3', 5'-GGAGAATTCTCGTCGTCCCAGAA-3') and BCMA-KO (5'-ATA AATGGCTACTGCACTTTCGGC-3', 5'-ACAAAGATAGTCCGTGGGTGTT TG-3') loci.

## DNA extraction and PCR

DNA from mouse biopsies was extracted using the SampleIn Direct PCR Kit (highQu, Cat.: DPK0101) and analyzed with the gene-specific primers. Primers used for genotyping BCMA wildtype (5'-GATCGGCTC

AGCTGGACAAG-3', 5'-CTTCACACCAGTTAGGAAGC-3'), BCMA:Tom (5'-GGACGAGCTGTACAAGTGATG-3', 5'-TTGGTTGCCCTGGAACTAGC-3'), and BCMA-KOΔ3 (5'-CCGCATAACTTCCAAGAGCC-3', 5'-CTCCGA ACAATTACACACTTCATAGT-3').

## Immunizations

The 10–30-week-old BCMA-KOΔ3 or BCMA-KO and wildtype control mice were immunized intraperitoneally with 100 μg

**Fig. 4 | Flow cytometric analysis of antigen-specific ASCs and ASC subsets in BCMA-KOΔ3 mice upon prime/boost immunization with the mRNA vaccine mRNA-1273. A** Schematic illustration of the experimental setup. Mice were intramuscularly immunized and boosted on day 42 with 5 μg mRNA-1273 each and analyzed on day 126. **B** RBD-specific IgG serum concentrations were determined by ELISA for wildtype (WT, black, $n = 7$) and BCMA-KOΔ3 mice (KOΔ3, gray, $n = 9$). Flow cytometric quantification of (**C**) RBD-specific CD138$^+$TACI$^+$ ASCs in the bone marrow ($n = 7$ WT and 8 KOΔ3) and (**D**) total ASC numbers in bone marrow, spleen, and mLN ($n = 7$ WT and 9 KOΔ3). **E** IgH isotype-specific total and RBD-specific ASC counts in the bone marrow were determined by flow cytometry and visualized in a stacked bar diagram. **F** Concentration of total serum IgM, IgG, and IgA and feces IgA in BCMA-KOΔ3 and wildtype mice at day 126 ($n = 8$ WT and 9 KOΔ3). Bar diagrams

show mean and SD with each dot indicating one mouse. Statistical analysis in B and E was performed using a two-way ANOVA with Šídák's multiple comparisons test. Statistical analysis (**C**, **D**, **F**) was performed with unpaired $t$-tests. Exact $p$-values, mouse sex, and ages are provided in the Source Data file. **G**, **H** Transcriptome analysis of bone marrow ASCs isolated of mRNA-1273 immunized BCMA-KOΔ3 and wildtype mice (WT) on day 126 after primary immunization. Principal component analysis visualizes sample similarities (**G**), and differential gene expression is documented in the MA plot (**H**). In BCMA-KOΔ3 ASCs, no upregulated genes were detected; the only downregulated gene (*Tnfrsf17*) is colored in blue. ASC antibody-secreting cells, RBD receptor binding domain of the SARS-CoV-2, i.m. intramuscular, ns not significant, mLN mesenteric lymph node, $^*p \leq 0.05$, $^{**}p \leq 0.01$, $^{****}p \leq 0.0001$.

(100 μl PBS) NP$_{(20)}$-KLH (LGC Biosearch Technologies, Cat.: N-5060-25) in 100 μl alum. Mice were euthanized on day 49 or boosted on day 42, with 50 μg NP$_{(20)}$-KLH in a total volume of 200 μl PBS. For mRNA-1273 immunization, 10–20-week-old mice were immunized and boosted intramuscularly into both hind legs after 42 days with 25 μl mRNA-1273 vaccine diluted in 25 μl sterile PBS at each time point. Blood samples were taken by puncturing the vena facialis or by cardiac puncture of euthanized mice at the end of an experiment.

## Flow cytometry

Single-cell suspensions for flow cytometric analyses were prepared as described[42]. Briefly, cell suspensions were depleted of red blood cells (RBC) (RBC lysis buffer, BioLegend, Cat.: 420301) and incubated with Fc block (anti-mouse CD16/CD32, clone 93, ThermoFisher, Cat.: 14-0161-82) for 5 min at RT and then stained with various combinations of the following antibodies: CD138-PE.Cy7 (clone 281-2, BioLegend, Cat.: 142514, 1:1500), TACI-PE (clone eBio8F10-3, ebioscience, Cat.: 12-5942, 1:200), TACI-APC (clone eBio8F10-3, ebioscience, Cat.: 17-5942, 1:400), TACI-BV421 (clone 8F10, BD, Cat.: 742840, 1:600 B220-PerCPCy5.5 (clone Ra3-6b2, ebioscience, Cat.: 45-0452-80, 1:200), CD19-BV421 (clone 6D5, BioLegend, Cat.: 115538, 1:200), IgA-FITC (polyclonal, Southern Biotech, Cat.: 1040-02, 1:1000), IgA-AF647 (polyclonal, Southern Biotech, Cat.: 1040-31, 1:10,000), IgM-Biotin (polyclonal, Jackson, Cat.: 115-065-075, 1:1000), anti-human IgG-AF647 (polyclonal, Southern Biotech, Cat.: 2048-31, 1:1000), CD38-APC.Cy7 (clone 90, BioLegend, Cat.: 102727, 1:1000), GL7-FITC (clone GL7, BD, Cat.: 562080, 1:100), Ki-67-APC (clone 16A8, BioLegend, Cat.: 652405, 1:150), NP$_{(28)}$-PE (Biosearch, Cat.: N-5070-1, 1:200), Streptavidin (SAV)-FITC (ebioscience, Cat.: 11-4317-87, 1:500), Streptavidin-APC.Cy7 (biolegend, Cat.: 405208, 1:1600), CD19-APCFire750 (clone 6D5, BioLegend, Cat.: 115558, 1:400), CD38-PerCPCy5.5 (clone 90, BioLegend, Cat.: 102722, 1:100), and RBD-biotin (BioLegend, Cat.: 793904). The anti-BCMA antibody (clone 25C2)[43] was produced and purified as described in ref. [10]. For staining of BCMA, RBC-depleted splenic single cell suspensions were seeded in R10 medium (RPMI-1640 supplemented with 1 mM sodium pyruvate, 2 mM L-glutamine, 100 U/ml penicillin-streptomycin, 50 μM β-mercapto-ethanol, 10% fetal calf serum (FCS)) at densities of $0.25 \times 10^6$ cells/ml in a humidified atmosphere at 37 °C with 5% CO$_2$ and incubated with the 1 μM γ-secretase inhibitor DAPT (Calbiochem Merck, Cat.: D5942) or dimethyl sulfoxide (DMSO) as solvent control for 18 h[18]. For intracellular staining of Ki-67 or RBD, cells were fixed and permeabilized using the Fix & Perm Kit by Nordic-MUbio (Cat.: GAS-002) according to the manufacturer's instructions. Biotinylated RBD protein and FITC, and APC.Cy7 coupled SAV antibodies were diluted 1:100 in FACS buffer, each in a separate tube. The dilution of biotinylated RBD protein was mixed 1:1 with either SAV-FITC or SAV-APC.Cy7 and incubated on ice for 30 min, protected from light to allow the formation of RBD-SAV complexes. FITC and APC.Cy7 coupled RBD-SAV complexes were combined just prior to intracellular staining of the cells.

Samples were analyzed with a Gallios flow cytometer (Beckman Coulter), and data were evaluated using the Kaluza Analysis software (Beckman Coulter, version 2.2). The full gating strategy is described in the supplementary material (Fig. S5).

## RNA sequencing and analysis

ASCs (CD138$^+$ TACI$^+$) were isolated on a BeckmanCoulter MoFlo Astrios EQ from bone marrow or spleen single-cell suspensions of the untreated or immunized BCMA-KOΔ3 and wildtype mice and sorted directly into RLT lysis buffer (Qiagen). Total RNA (RNeasy microKit, Qiagen, Cat.: 74004) from the isolated ASCs was used to prepare sequencing libraries using the Clontech SMART-Seq v4 kit. These libraries were prepared and sequenced on an Illumina HiSeq X instrument (2 × 150bp) by Admera Health LLC. The reads were aligned to the mouse reference genome (GRCm38.p6) using STAR (v2.7.10)[44], and gene-wise counts were generated with salmon (v1.10.01)[45]. Differential expression analysis was conducted using the R (v4.3) package edgeR (v4.0.0)[46]. Genes with low expression were excluded with the "filterByExpr" function, and immunoglobulin sequences were removed from the analysis. Libraries were normalized with the "TMM" method before testing for differential expression between the BCMA-KOΔ3 and wildtype samples with the "exactTest" function. Genes with a fold change >1.5 and a false discovery rate (FDR) ≤ 0.05 were determined as significant.

## ELISA

Blood was transferred into BD microtainer© blood collection tubes, incubated for 30 min at room temperature (RT), and centrifuged for 90 s at full speed at RT. Feces samples were collected at the end of an experiment, dissolved in 100 μl PBS/mg feces, and centrifuged at maximum speed for 5 min in a tabletop centrifuge (Eppendorf centrifuge 5424), and the supernatant was collected and used as a feces sample, further diluted at 1:100 in PBS-2% FCS. Sera were diluted as follows: total IgG 1:10,000, total IgM/IgA 1:4000. For Antigen-specific detection, sera were diluted 1:250 and 1:500 starting day 21 after immunization, and 1:2000 for all time points after boost. For detecting serum or feces Ig by ELISA, 96-well flat bottom plates were coated with 50 μl/well of a 1 μg/ml solution with goat anti-mouse IgM, IgG or IgA (SouthernBiotech, Cat.: 1021-01, 1030-01, 1040-01), or for detection of antigen-specific Ig with 50 μl/well of a 1 μg/ml solution with NP$_{(20)}$-bovine serum albumin (BSA) (Biosearch Technologies, Cat.: N-5050H10) or 400 ng/ml SARS-CoV-2 RBD in ELISA coating buffer (15 mM Na$_2$CO$_3$ and 35 mM NaHCO$_3$ in dH$_2$O). Unspecific binding was blocked with PBS-2% FCS for 1 h at RT. Sera or feces supernatant dilutions in PBS-2% FCS were incubated at 4 °C overnight or at RT for 2 h. As detection antibodies, 50 μl/well HRP-coupled goat-anti-mouse IgM (0.3 μg/ml), IgG (1 μg/ml), or IgA (0.2 μg/ml) (Southern Biotech, Cat.: 1021-01, 1030-01, 1040-01) were incubated for 1 h at RT. The TMB Substrate Reagent Set (BC OptEIA™, Cat.: 555214) was used following the manufacturer's protocol. ELISA plates were measured and analyzed using the Biolegend Mini ELISA plate reader at 450 nm. Analysis was performed using the *"Four Parameter Logistic Curve"* online data

*analysis tool, MyAssays Ltd., accessed in the time of 2021–2024*, http://www.myassays.com/four-parameter-logistic-curve.assay.

## ELISpot analysis

ASCs were quantified in bone marrow and splenic single-cell suspensions by ELISpot analysis in 96-well flat-bottom plates as described in refs. 47,48. To analyze total or NP-specific ASCs, the plates were coated as described for ELISA with 2 µg/ml of the respective antibody. Alkaline phosphatase (AP)-coupled goat-*anti*-mouse IgG, IgM, or IgA (Southern Biotech., Cat.: 1021-04, 1030-04, 1040-04) were used as detection antibodies with 50 µl/well at a concentration of 0.25 µg/ml. 50 µl/well of ESA substrate solution containing 5-Bromo-4-chloro-3-indolyl phosphate p-toluidine salt (BCIP, Sigma-Aldrich, Cat.: B8503) was used for detection (ESA substrate buffer 10×: 100 ml 1.5 M AMP pH 10.3; 0.75 ml 1 M MgCl2; 0.15 ml Triton X-405 70%; 1.5 ml NaN3 10%; 47.6 ml H$_2$O; adjust to pH 10.25 with HCl; filter and store at 4 °C protected from light. ESA substrate solution: 50 ml 10× ESA-substrate buffer; 500 mg BCIP; 450 ml H$_2$O; stir for 1 h at room temperature protected from light; filter and store at 4 °C protected from light. Spots representing single ASCs were counted using the ImmunoSpotR© Series 6 Ultra-V Analyzer from C.T.L. and analyzed with the C.T.L. Software BioSpotR© ImmunoSpot (v5.1.36).

## Statistics

Statistical analyses were performed using Prism (GraphPad, v9.4). Prior to hypothesis testing, the normal distribution of values was assessed with the Shapiro–Wilk test. For nonparametric variables, the two-tailed Mann–Whitney test was used. For parametric variables, two-tailed unpaired *t*-tests, correcting for multiple comparisons by the FDR according to Benjamini, Krieger, and Yekutieli, the two-stage step-up method was performed. Multiple comparisons were performed using a two-way ANOVA with Šídák's multiple comparisons test. A maximum of one outlier per group was identified by Grubb's outlier test and excluded from analysis. Numerical values are given by mean and SD. A *p*-value ≤ 0.05 was considered significant.

## Reporting summary

Further information on research design is available in the Nature Portfolio Reporting Summary linked to this article.

## Data availability

The RNA-Seq data generated during this study are available at GEO: GSE277098 (https://www.ncbi.nlm.nih.gov/geo/query/acc.cgi?acc=gse277098). All data underlying graphs and quantitative analysis are provided in the Source Data File. Source data are provided with this paper.

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

## Acknowledgements

We thank Heidi von Berg for expert animal care, Manuela Hauke for purifying the 25C2 antibody, Leonie Somann for performing preliminary experiments, Uwe Appelt and Markus Mrotz for cell sorting in the Core Unit "Cell Sorting and Immunomonitoring" (Friedrich-Alexander University), and the University Hospital Erlangen pharmacy for providing leftover doses of the mRNA-1273 (Spikevax) vaccine. The work was supported in part by the German Research Foundation (DFG) through project grants TRR130, GRK1660, and GRK2599, from the Federal Ministry of Education and Research (BMBF) through the "NaFoUniMed-Covid19 " (FKZ: 01KX2021)—COVIM consortium, the Kastner foundation, and the "Interdisziplinäre Zentrum für Klinische Forschung (IZKF)" of the FAU Erlangen-Nürnberg.

## Author contributions

H.-M.J. conceived the project, and S.R.M., K.P., S.R.S., and H.-M.J. designed the experiments. S.R.M., K.P., and S.R.S. performed and analyzed the experiments. S.R.S. analyzed bioinformatics data. S.B. and T.H. W. generated the BCMA-KO mouse. J.W., E.R., L.W., J.T., and W.S. assisted in performing the experiments. S.R.M., S.R.S., and H.-M.J. wrote the manuscript, and all authors revised it.

## Funding

## Competing interests

The authors declare no competing interests.
