## [Transparent Peer Review file · Nature Communications]

B cell maturation antigen (BCMA) is dispensable for the survival of long-lived plasma cells

Corresponding Author: Professor Hans-Martin Jack

Version 0:

Reviewer comments:

Reviewer #1

(Remarks to the Author)

This manuscript tackles an important dogma in the field of plasma cell biology. More precisely, Menzel et al. challenge the original view that APRIL pro-survival effect on plasma cells relies on the receptor BCMA. Indeed, previous work published 20 years ago suggested that plasma cell maintenance in the bone marrow was reduced in BCMA deficient mice. Since then, the role of this receptor in plasma cell survival had not really been confirmed.

One technical difficulty arises from the low surface expression of BCMA on plasma cells as the receptor is rapidly shed by gamma-secretase, making it difficult to interrogate which plasma cells actually express BCMA.

The authors generated a BCMA reporter mice to study this question and showed that, in adult mice, BCMA is apparently uniquely expressed by mature plasma cells (associated manuscript).

By crossing their reporter model with an E2A-cre mice they generated a new BCMA deficient mouse model. These mice display a normal plasma cell compartment in the spleen and bone marrow at steady state and after a primary immunization suggesting that BCMA is actually dispensable for maintenance of the plasma cell pool. This was confirmed at steady state in a second and independent BCMA deficient model. In addition, the authors showed that, after a secondary immunization, total plasma cell numbers almost double in BCMA deficient animals compared to WT control. This surprising result suggests that BCMA may actually act as a negative regulator of the total plasma cell pool by blocking APRIL through its soluble cleaved form.

Altogether this is a very solid study, conducted using state of the art techniques and appropriate controls. The main message of the paper might be negative but it is an important one that will open new lines of research to better understand plasma cell long-term maintenance, a key question to improve antibody-mediated vaccine response.

I only have few comments:

-The authors analyze the plasma cells at the population level. With their current experimental setting they cannot discriminate whether the proliferation or the turnover rate of individual plasma cells are affected. Indeed, a decreased survival rate might be counterbalanced by an increased proliferation for example. To exclude a change in the turnover rate of plasma cells within the bone marrow and clearly demonstrate that BCMA is dispensable for long-lived plasma cell survival, the authors could perform a pulse of EdU or BrdU labelling followed by long-term follow up with several time points of the labelled plasma cells in BCMA deficient and control mice.

-Is there any impact of BCMA deficiency on the maintenance of high affinity plasma cells? This could be looked at by measuring the ratio of antibodies binding to low valency/high valency NP in the serum or by ELISpot.

-Circulating IgA and mesenteric lymph node total plasma cells are increased at steady state. This should at least be mentioned in the text and discussed with the results of the memory immunization protocols.

-The rise of bystander IgA, and to a lesser extent, IgM plasma cells after a boost immunization in BCMA deficient mice is really intriguing. Was it also observed in the second BCMA deficient model? Although a bit out of the scope it would be really interesting to measure APRIL bioavailability in bone marrow and/or lymph node fluids of WT and BCMA deficient mice. If not feasible, the potential role of shed BCMA as an APRIL "scavenger" should be discussed.

Minor points:

Figure 1E: The “downregulated genes colored in blue” are not really visible on the figure. Also, as there are not so many genes, their name could be indicated on the volcano plot for total transparency.

Figure 4E: The organ analyzed is not indicated. I guess it is the bone marrow?

Statistics: It is indicated in the legend of supplementary figure 1 that gaussian distribution was tested to determine the use of a t-test or a Mann-Whitney test. The method used to assess the gaussian distribution should be indicated in the main methods.

Reviewer #2

(Remarks to the Author)

The manuscript by Menzel et al. revisits the role of BCMA as a critical survival receptor for long-lived plasma cells in mice. This study was prompted by conflicting findings from two seminal *in vivo* studies using BCMA KO mice: O'Connor et al. (JEM, 2004) demonstrated that BCMA is required for the survival of long-lived bone marrow plasma cells, whereas Xu and Lam (Mol Cell Biol, 2001) found that BCMA KO mice mount normal primary and secondary immune responses to T cell-dependent antigens. To address this discrepancy, Menzel et al. generated two independent BCMA KO strains: one with exon 3 deleted and another with a full-length BCMA deletion. Across multiple experimental conditions – including non-immunized mice, mice immunized with a foreign TD antigen, and mice vaccinated with the SARS CoV-2 mRNA vaccine – the authors observed no significant differences in total or antigen-specific plasma cell numbers in the spleen or bone marrow, nor in serum antibody titers, compared to controls. Secondary immune responses were also comparable, with similar antigen-specific PC frequencies in the bone marrow.

This study employs rigorous experimental approaches, methodologies, and controls to evaluate BCMA deficiency and plasma cell frequencies. The findings from the data support the conclusion that BCMA is dispensable for long-lived PC maintenance. These findings confirm Xu and Lam's findings that BCMA exon 3 deficiency does not affect splenic PC numbers, basal serum Ig levels, antigen-specific IgM and IgG titers, or germinal center responses. Notably, Menzel et al. extend previous studies by comprehensively evaluating bone marrow PCs and their IgM/IgA/IgG isotypes. However, beyond confirming these prior observations, the study does not provide new mechanistic insights into bone marrow plasma cell survival in the absence of BCMA beyond those considered in the O'Connor and Xu and Lam studies.

A key distinction between these earlier studies is the presence or absence of TACI expression in PCs. As noted by Xu and Lam, TACI may provide sufficient survival signals to PCs when BCMA is absent. Additionally, elevated APRIL levels in BCMA-deficient mice could mediate PC survival via TACI, as suggested by Jiang et al. (J Immunol, 2011). Benson et al. (JI, 2008) further demonstrated that APRIL, alone or with BAFF, supports PC survival, as shown by treating immunized WT and APRIL KO mice with TACI-Ig or BAFF-Ig. This could explain PC survival in BCMA-KOdelta3 mice, where TACI expression was used for PC identification and sorting for transcriptomics. To compare PC transcriptomic profiles between BCMA-KOdelta3 and WT mice, Menzel et al. analyzed bone marrow PCs from non-immunized and mRNA-immunized mice, identifying *Tnfrsf17* as the sole differentially expressed gene. However, the absence of differences in survival gene expression could be attributed to persistent APRIL-TACI signaling, independent of BCMA. Notably, the study does not assess PC responses to direct APRIL stimulation, leaving unresolved the question of whether alternative environmental signals support long-lived PC survival in the absence of BCMA. The potential significance of this would have a greater impact on moving the field forward.

Minor comments:

1. Gamma secretase inhibitor treatment should be included or cited in M&M.
2. How the results of this study apply to the survival of human PCs should be discussed.

Reviewer #3

(Remarks to the Author)

Menzel et al. report the effect of BCMA deletion on antibody producing plasma cells (ASCs) *in vivo*, either with or without immunization with T-dependent antigens (NP-KLH and SARS-COV2 mRNA). They used mostly a knock-out model, which deletes BCMA exon 3 encoding cytoplasmic signaling domain, similar to previously published models that resulted in inconsistent results. Furthermore, they corroborated some of their findings with a second knock out model deleting the whole BCMA locus. They do not see any decrease in total or antigen-specific IgM+, IgG+ or IgA+ ASCs in bone marrow, spleen or mesenteric lymph nodes, nor any decreased in serum IgG, IgA and IgM in these models, either with or without primary and secondary immunization with TD antigens. This is in contrast with one previous paper that suggested decreased antigen specific IgG+ ASCs in BM following immunization, but in agreement with another publication using a similar exon3 knock out model. The authors therefore demonstrate that BCMA is dispensable for long-lived plasma cell generation and survival, at least under the experimental conditions analyzed.

Overall, the experiments are well performed and the data are convincing and clear and settle the previous controversy about the requirement for BCMA in the generation and maintenance of long-lived ASCs.

Major comments:

1. BCMA has been strongly implicated in the generation of IgA+ ASCs in the gut and evidence suggests that activation and differentiation of IgA+ B cells in the gut may differ according to type of antigen and immunization route. A major advance in the fuller understanding of the role of BCMA would be gained by performing oral immunization in BCMA-/- mice and controls, for example using microbial antigens, and investigating IgA+ ASCs in gut tissue.

2. Line 281-284: "Given that the cleaved extracellular fragment of BCMA binds and masks the pro-survival factor APRIL, this APRIL-decoy function may be the primary role of murine BCMA. Without this soluble decoy, increased availability of APRIL in the ASC microenvironments may increase the survival probabilities and longevity of ASCs in the bone marrow".

The authors suggest that BCMA is a decoy receptor based on the increased total IgM and IgA ASCs after boost immunization. BCMA is shed directly by gamma-secretase at steady state and the soluble form of BCMA is indeed known to be a decoy receptor for APRIL/BAFF (Laurent et al. Nat Com. 2015). However, the membrane-bound form, albeit expressed at low levels, is a signaling molecule that can activate NFkB basally and more so following ligand binding. Given the complexity of the BCMA/TACI/APRIL/BAFF system and redundancies of these molecules, I think that the data presented are not sufficient to state that the main function of BCMA (membrane and soluble forms) is a decoy receptor. Indeed Eslami et al. have shown that BCMA knock out leads to upregulation of TACI and that TACI and BCMA, as well as BAFF and APRIL, have redundant functions in ASC differentiation and survival. Do the authors also detect an upregulated on TACI in their BCMA-/- B cells and ASCs? It would also have been interesting to investigate NFkB signalling in BCMA+ and KO ASCs, in presence or absence of APRIL and BAFF stimulation.

Minor points:

- a) The authors state that DN cells are mostly IgG+ ASCs. The data to support this should be shown.
- b) Line 179: Decrease, not decreased.

Version 1:

Reviewer comments:

Reviewer #1

(Remarks to the Author)

The authors have adequately addressed most of my comments experimentally or by discussing their results. I have no further comments.

Reviewer #2

(Remarks to the Author)

No additional comments of the revised manuscript.

Reviewer #3

(Remarks to the Author)

The authors have provided additional data and have modified the manuscript in response to the reviewers comments. My view is that they have adequately replied to the queries raised. The methodology is sound and the results settle previous controversies about the function of BCMA, an important target for immunotherapy.

Comments to the Reviewers

Menzel et al.

Note: Reviewer comments are in BLACK, and authors' responses are in RED and ITALICS

Reviewer #1

This manuscript tackles an important dogma in the field of plasma cell biology. More precisely, Menzel et al. challenge the original view that APRIL pro-survival effect on plasma cells relies on the receptor BCMA. Indeed, previous work published 20 years ago suggested that plasma cell maintenance in the bone marrow was reduced in BCMA deficient mice. Since then, the role of this receptor in plasma cell survival had not really been confirmed.

One technical difficulty arises from the low surface expression of BCMA on plasma cells as the receptor is rapidly shed by gamma-secretase, making it difficult to interrogate which plasma cells actually express BCMA.

The authors generated a BCMA reporter mice to study this question and showed that, in adult mice, BCMA is apparently uniquely expressed by mature plasma cells (associated manuscript).

By crossing their reporter model with an E2A-cre mice they generated a new BCMA deficient mouse model. These mice display a normal plasma cell compartment in the spleen and bone marrow at steady state and after a primary immunization suggesting that BCMA is actually dispensable for maintenance of the plasma cell pool. This was confirmed at steady state in a second and independent BCMA deficient model. In addition, the authors showed that, after a secondary immunization, total plasma cell numbers almost double in BCMA deficient animals compared to WT control. This surprising result suggests that BCMA may actually act as a negative regulator of the total plasma cell pool by blocking APRIL through its soluble cleaved form.

Altogether this is a very solid study, conducted using state of the art techniques and appropriate controls. The main message of the paper might be negative but it is an important one that will open new lines of research to better understand plasma cell long-term maintenance, a key question to improve antibody-mediated vaccine response.

I only have few **comments**:

The authors analyze the plasma cells at the population level. With their current experimental setting they cannot discriminate whether the proliferation or the turnover rate of individual plasma cells are affected. Indeed, a decreased survival rate might be counterbalanced by an increased proliferation for example. To exclude a change in

the turnover rate of plasma cells within the bone marrow and clearly demonstrate that BCMA is dispensable for long-lived plasma cell survival, the authors could perform a pulse of EdU or BrdU labelling followed by long-term follow up with several time points of the labelled plasma cells in BCMA deficient and control mice.

*We thank the reviewer for the very positive overall assessment and constructive comments. We agree that a direct assessment of plasma cell turnover using EdU/BrdU pulse-chase experiments might provide additional insight into survival/turnover dynamics. However, our current approach—which includes comprehensive quantification of antigen-specific and total ASC numbers over time as well as **transcriptomic analyses showing no changes in survival-related gene expression**—strongly supports our main conclusion that BCMA deficiency does not compromise long-lived plasma cell survival. In addition, **flow cytometric analyses of the proliferation marker Ki67** in bone marrow ASC of BCMA-WT and KOΔ3 mice did not reveal significant differences between genotypes, arguing against the possibility that ASC in BCMA-deficient mice were more proliferative (data not shown in the manuscript due to figure limitations). Considering these converging data, we believe that performing an extensive pulse-chase experiment is not essential to support our conclusions and would unnecessarily extend the revision timeline, as we would face the lengthy German application process for animal experiments. This would take at least 6 months until approval, in addition to the time needed to perform the experiment, which would take approximately an additional 3 months. **We have included a discussion of this point in the revised Discussion section.***

Figure: Flow cytometric analysis of Ki-67 protein abundance in bone marrow and splenic ASCs from BCMA-KOΔ3 and wildtype mice. Mice were analyzed either unimmunized (left panel) or 7 weeks after primary immunization with NP-KLH in Alum.

Is there any impact of BCMA deficiency on the maintenance of high affinity plasma cells? This could be looked at by measuring the ratio of antibodies binding to low valency/high valency NP in the serum or by ELISpot.

*We agree that this aspect is intriguing. We, therefore, investigated the impact of BCMA deficiency on the maintenance of high affinity plasma cells by measuring the ratio of IgG binding to NP₄/NP₂₀ in the serum at day 49 (primary immunization of BCMA-KO mice, **Supplementary Figure S2F**) and at day 128 (after a boost, BCMA-KOΔ3, **Supplementary Figure S2G**). The binding affinity of IgG was comparable between WT and BCMA-deficient mice, suggesting that BCMA does not impact the maintenance of high-affinity plasma cells. We have included these results in the **revised Results section on page 7 and 8**.*

Circulating IgA and mesenteric lymph node total plasma cells are increased at steady state. This should at least be mentioned in the text and discussed with the results of the memory immunization protocols.

*In the second BCMA-KO mouse model, we did not see an increase in circulating IgA (non-immunized mice, **Supplementary Figure S2C**) and mLN ASCs (after primary immunization with NP-KLH, data not shown). However, we now mention the increased abundance of serum IgA and IgA+ ASCs in mLN of BCMA-KOΔ3 mice in the **Results section** and have included a discussion of this point in the **revised Discussion section on page 13**.*

The rise of bystander IgA, and to a lesser extent, IgM plasma cells after a boost immunization in BCMA deficient mice is really intriguing. Was it also observed in the second BCMA deficient model?

*To reproduce the experimental setup by O'Connor et al. and to confirm our contradicting observation that ASC survival was not affected in BCMA-KOΔ3 mice after primary immunization, mice from a second line with the entire deletion of all three BCMA exons were immunized only with the primary immunization regimen with NP-KLH. After primary immunization, we did not detect an increase of the ASC population in either mouse model after 7 weeks (**Figure 2D** and data not shown). However, as the increase of ASC numbers after boost was reproducible with a second antigen in the BCMA-KOΔ3 mice, we did not feel it would add meaningful information to repeat the prime/boost immunization in the second BCMA-KO mouse model.*

Although a bit out of the scope it would be really interesting to measure APRIL bioavailability in bone marrow and/or lymph node fluids of WT and BCMA deficient mice. If not feasible, the potential role of shed BCMA as an APRIL "scavenger" should be discussed.

*We fully agree that direct measurements of APRIL abundance in BCMA-WT and BCMA-KO mice would be informative in addressing BCMA's function as a soluble APRIL scavenger. We attempted to assess APRIL abundances in the sera and freshly isolated bone marrow both before and after immunization. However, we could not reliably quantify APRIL due to the insufficient assay sensitivity and performance of available commercial mouse APRIL detection assays (data not shown), which was also noted in another publication (Miao et al. 2022, *JExpMed* and PhD thesis, Thang, 2023, "Neutralization of BAFF and APRIL with engineered soluble BCMA decoy receptor for the treatment of B cell malignancies"). These limitations could not be overcome by extensive optimization of in-house developed competitive- and*

sandwich-ELISA approaches using tagged APRIL variants or BCMA-Ig together with a reportedly non-competing APRIL-binding monoclonal antibody derived from patent literature (mAb clone 3530 from patent WO2017091683A1). Another challenge in detecting APRIL in the bone marrow niche arises from the possibility that APRIL is sequestered on cell surfaces, e.g., by proteoglycans and is thus not present in the isolated bone marrow supernatants. The use of state-of-the-art techniques, e.g., NULISA to detect trace amounts of proteins is not possible due to lack of APRIL as a target in the provided commercial mouse Kits. We are, therefore, technically constrained from verifying the prediction of increased free APRIL that would increase from the ablation of soluble BCMA acting as an APRIL-decoy receptor in the blood or the survival niches.

Minor points:

figure 1E: The "downregulated genes colored in blue" are not really visible on the figure. Also, as there are not so many genes, their name could be indicated on the volcano plot for total transparency.

*Differential gene expression analysis yielded only one downregulated gene, namely Tnfrsf17. This gene is already highlighted in blue. To avoid confusion, we have **changed the figure legend of Fig. 1** to: "..., and the only downregulated gene is colored in blue."*

Figure 4E: The organ analyzed is not indicated. I guess it is the bone marrow?

*We apologize for the missing information. We have **indicated the analyzed organ in Figure 4E** and adapted the respective figure legend.*

Statistics: It is indicated in the legend of supplementary figure 1 that gaussian distribution was tested to determine the use of a t-test or a Mann-Whitney test. The method used to assess the gaussian distribution should be indicated in the main methods.

We apologize for the omission of the requested information. We have included this information in the Methods section.

Reviewer #2

The manuscript by Menzel et al. revisits the role of BCMA as a critical survival receptor for long-lived plasma cells in mice. This study was prompted by conflicting findings from two seminal *in vivo* studies using BCMA KO mice: O'Connor et al. (JEM, 2004) demonstrated that BCMA is required for the survival of long-lived bone marrow plasma cells, whereas Xu and Lam (Mol Cell Biol, 2001) found that BCMA KO mice mount normal primary and secondary immune responses to T cell-dependent antigens. To address this discrepancy, Menzel et al. generated two independent BCMA KO strains: one with exon 3 deleted and another with a full-length BCMA deletion. Across multiple experimental conditions – including non-immunized mice, mice immunized with a foreign TD antigen, and mice vaccinated with the SARS CoV-2 mRNA vaccine – the authors observed no significant differences in total or antigen-specific plasma cell numbers in the spleen or bone marrow, nor in serum antibody titers, compared to controls. Secondary immune responses were also comparable, with similar antigen-specific PC frequencies in the bone marrow.

This study employs rigorous experimental approaches, methodologies, and controls to evaluate BCMA deficiency and plasma cell frequencies. The findings from the data support the conclusion that BCMA is dispensable for long-lived PC maintenance. These findings confirm Xu and Lam's findings that BCMA exon 3 deficiency does not affect splenic PC numbers, basal serum Ig levels, antigen-specific IgM and IgG titers, or germinal center responses. Notably, Menzel et al. extend previous studies by comprehensively evaluating bone marrow PCs and their IgM/IgA/IgG isotypes.

However, beyond confirming these prior observations, the study does not provide new mechanistic insights into bone marrow plasma cell survival in the absence of BCMA beyond those considered in the O'Connor and Xu and Lam studies.

A key distinction between these earlier studies is the presence or absence of TACI expression in PCs. As noted by Xu and Lam, TACI may provide sufficient survival signals to PCs when BCMA is absent. Additionally, elevated APRIL levels in BCMA-deficient mice could mediate PC survival via TACI, as suggested by Jiang et al. (J Immunol, 2011). Benson et al. (JI, 2008) further demonstrated that APRIL, alone or with BAFF, supports PC survival, as shown by treating immunized WT and APRIL KO mice with TACI-Ig or BAFFR-Ig. This could explain PC survival in BCMA-KO Δ 3 mice, where TACI expression was used for PC identification and sorting for transcriptomics.

To compare PC transcriptomic profiles between BCMA-KO Δ 3 and WT mice, Menzel et al. analyzed bone marrow PCs from non-immunized and mRNA-immunized mice, identifying *Tnfrsf17* as the sole differentially expressed gene. However, the absence of differences in survival gene expression could be attributed to persistent APRIL-TACI signaling, independent of BCMA.

Notably, the study does not assess PC responses to direct APRIL stimulation, leaving unresolved the question of whether alternative environmental signals support long-

lived PC survival in the absence of BCMA. The potential significance of this would have a greater impact on moving the field forward.

We appreciate the reviewer's detailed feedback and the opportunity to clarify key aspects of our study. While we recognize that the studies by Xu and Lam (2001) and O'Connor et al. (2004) have been interpreted as conflicting, we note that both reported unaltered initiation of T cell-dependent humoral immune responses. The key difference lies in the time points analyzed and the methods used: Xu and Lam observed sustained antigen-specific serum titers (by ELISA), while O'Connor et al. reported a loss of antigen-specific ASCs (by ELISpot). O'Connor et al. based their reported conclusion on the importance of the APRIL-BCMA axis in large parts on the proposition that the second APRIL-receptor TACI is only expressed at low levels in plasma cells. This assumption has been clearly refuted, with more recent work by our lab and numerous others demonstrating high TACI surface abundances in murine antibody-secreting cells. However, our study was not primarily motivated by reconciling these prior results but rather by the discovery of BCMA shedding. This aspect was not considered in earlier studies and could have influenced their interpretations.

We respectfully disagree with the notion that our study provides no new mechanistic insights. Indeed, in contrast to Peperzak et al. (2013), we find no reduction in bone marrow plasma cell numbers or reduced Mcl-1 mRNA transcripts in non-immunized BCMA-deficient mice. Therefore, albeit a negative but critical result for plasma cell survival, we conclude that membrane BCMA on plasma cells does not contribute to signaling networks that control the longevity of plasma cells and antibody responses. This is a crucial mechanistic insight because, as mentioned by reviewer 1, our results "open new lines of research to better understand long-term maintenance of plasma cells."

Our data of unaltered antigen-specific plasma cell numbers and antibody titers up to day 128 after immunization in two independent BCMA-deficient mouse lines clearly supports the existence and importance of BCMA-independent pathways of ASC survival, as noted by the reviewer. As acknowledged by the reviewer, we did not detect upregulation of any compensatory survival pathways in the transcriptome analysis of BCMA-deficient ASCs. This strengthens our argument that BCMA as a membrane or soluble receptor is dispensable for long-lived plasma cell maintenance.

*Based on elevated TACI transcript abundances in total splenocytes from non-immunized wildtype (mixed genetic background) and BCMA-deficient mice (shown for n=1 with semi-quantitative RT-PCR), Xu and Lam proposed that increased APRIL-TACI signaling could compensate for BCMA loss. However, in this approach most TACI transcripts are likely derived from non-ASC B cell populations, including marginal zone B and B1 cells with elevated TACI expression (Sintes et al., 2017, Nat. Comm., DOI: 10.1038/s41467-017-01602-4). Our data obtained by flow cytometry (Supplementary Fig. S1E, S2A, S2H) and RNA sequencing of isolated ASCs (Fig. 1E, Fig. 4H), demonstrate that TACI is neither upregulated in unimmunized mice or in mice after primary immunization and after boost with mRNA-1273. While our approach did not uncover a novel mechanism of PC survival, it still advances mechanistic insights into the established network of APRIL and its receptors, reinforcing the notion that **TACI – despite its lower affinity – may be the prime mediator of APRIL-dependent effects in ASC**. This finding is particularly relevant in light of previous studies that showed that either BAFF or APRIL can sustain bone marrow ASCs (e.g., Benson et al., 2008, Eslami et al. 2024)*

However, the functional role of TACI in ASC is still not solved because all studies available from genomic TACI-KO models already display severe phenotypes in earlier B cell stages. Conditional deletion of TACI, specifically upon plasmablast differentiation remains to be explored. Overall, we appreciate the reviewer's thoughtful comments, which have helped us refine the framing of our findings. While our study does not uncover a novel survival mechanism, it provides crucial mechanistic insights into plasma cell maintenance, challenging prior assumptions about BCMA's role and highlighting avenues for further investigation. We have incorporated these clarifications and additional references noted by reviewer 2 into our revision.

Minor comments:

1. Gamma secretase inhibitor treatment should be included or cited in M&M.

We added the DAPT inhibitor treatment and the relevant citation to the Methods section of our revised manuscript.

2. How the results of this study apply to the survival of human PCs should be discussed.

*We have **added a new paragraph in the Discussion** to address the impact of our findings on human plasma cell biology.*

Reviewer #3

Menzel et al. report the effect of BCMA deletion on antibody producing plasma cells (ASCs) in vivo, either with or without immunization with T-dependent antigens (NP-KLH and SARS-COV2 mRNA). They used mostly a knock-out model, which deletes BCMA exon 3 encoding cytoplasmic signaling domain, similar to previously published models that resulted in inconsistent results. Furthermore, they corroborated some of their findings with a second knock out model deleting the whole BCMA locus. They do not see any decrease in total or antigen-specific IgM+, IgG+ or IgA+ ASCs in bone marrow, spleen or mesenteric lymph nodes, nor any decreased in serum IgG, IgA and IgM in these models, either with or without primary and secondary immunization with TD antigens. This is in contrast with one previous paper that suggested decreased antigen specific IgG+ ASCs in BM following immunization, but in agreement with another publication using a similar exon3 knock out model. The authors therefore demonstrate that BCMA is dispensable for long-lived plasma cell generation and survival, at least under the experimental conditions analyzed.

Overall, the experiments are well performed and the data are convincing and clear and settle the previous controversy about the requirement for BCMA in the generation and maintenance of long-lived ASCs.

Major comments:

1. BCMA has been strongly implicated in the generation of IgA+ ASCs in the gut and evidence suggests that activation and differentiation of IgA+ B cells in the gut may differ according to type of antigen and immunization route. A major advance in the fuller understanding of the role of BCMA would be gained by performing oral immunization in BCMA^{-/-} mice and controls, for example using microbial antigens, and investigating IgA+ ASCs in gut tissue.

We thank the reviewer for the very positive overall assessment and constructive comments. We agree with Reviewer 3 that the recommended experiments might shed some light on the function of BCMA in gut plasma cell biology. As a first indication of whether BCMA might affect gut plasma cells, we have performed immunofluorescence staining of IgA of the small intestine (SI) from non-immunized BCMA-deficient mice, yielding unaltered ASC numbers in the SI by co-staining with J chain to confirm plasma cell identity (Figure below, data not shown in manuscript).

Figure: Immunofluorescence analysis of IgA⁺ plasma cell numbers in the small intestine in BCMA-KOΔ3 mice. IgA-positive cells in the small intestine (SI) of BCMA-deficient and control mice were quantified in immunohistochemical staining with DAPI and anti-IgA antibody. For quantification, three independent people counted three SI sections per mouse in a double-blind, randomized way.

Further, we have quantified secreted IgA in the feces of mRNA-1273 immunized BCMA WT and KOΔ3 mice (Fig. 4F), showing no change in fecal IgA abundances. We are aware that the employed immunization protocols do not primarily induce mucosal immune reactions. Yet, they provide a first indication that the ASCs in the intestinal lamina propria do not display a dramatically different phenotype in a BCMA-deficient setting compared to their counterparts in the bone marrow and other secondary lymphoid tissues.

Therefore, while the suggested oral immunization is undoubtedly interesting, the outcome would not change our major message that BCMA is dispensable for plasma cell survival in the bone marrow. Additionally, oral immunization experiments are time-consuming mainly because the approval of a protocol to perform these additional immunizations by the State of Bavaria would take at least 6 months. This would significantly delay the timely publication of this manuscript to address a topic that we feel is peripheral to our key message. Therefore, while we acknowledge that the suggested experiments would be of interest, we consider them to be beyond the scope of our manuscript.

2. Line 281-284: "Given that the cleaved extracellular fragment of BCMA binds and masks the pro-survival factor APRIL, this APRIL-decoy function may be the primary role of murine BCMA. Without this soluble decoy, increased availability of APRIL in the ASC microenvironments may increase the survival probabilities and longevity of ASCs in the bone marrow".

The authors suggest that BCMA is a decoy receptor based on the increased total IgM and IgA ASCs after boost immunization. BCMA is shed directly by gamma-secretase at steady state and the soluble form of BCMA is indeed known to be a decoy receptor for APRIL/BAFF (Laurent et al. Nat Com. 2015). However, the membrane-bound form, albeit expressed at low levels, is a signaling molecule that can activate NFκB basally

and more so following ligand binding. Given the complexity of the BCMA/TACI/APRIL/BAFF system and redundancies of these molecules, I think that the data presented are not sufficient to state that the main function of BCMA (membrane and soluble forms) is a decoy receptor.

We agree with the reviewer that our data does not allow us to draw the conclusion that BCMA acts solely as a soluble decoy receptor. Therefore, we suggested that BCMA's "APRIL-decoy function may be the primary role of murine BCMA" (line 284).

Indeed Eslami et al. have shown that BCMA knock out leads to upregulation of TACI and that TACI and BCMA, as well as BAFF and APRIL, have redundant functions in ASC differentiation and survival. Do the authors also detect an upregulated on TACI in their BCMA^{-/-} B cells and ASCs?

*While the data shown by Eslami et al. imply a compensatory role of TACI in the absence of BCMA, we did not find evidence of elevated TACI surface abundances in this study. Our flow cytometric analysis of non-immunized BCMA-KO Δ 3 mice showed no upregulation of TACI on ASCs (**Supplementary Fig. S1E**), which was confirmed by transcriptomic data showing BCMA as the only differentially expressed gene (Fig. 1E). In line with an unaltered ASC compartment after primary immunization, an increased TACI abundance was not seen after primary immunization with NP-KLH in both BCMA-deficient mouse models (**Supplementary Fig. S2A, H**). After prime-boost immunizations, we found increased ASC numbers in the bone marrow and mLN. However, increased TACI abundance could not be detected after prime-boost immunization with mRNA-1273 (data not shown), further supported by the ASC transcriptome analysis (Fig. 4H), excluding TACI upregulation as a general survival mechanism to compensate BCMA-deficiency.*

We agree with Eslami et al. that TACI and BCMA could have redundant roles in the absence of the other receptor. Yet, under physiological conditions with both receptors present, we think it is highly unlikely that BCMA functions primarily as a signaling molecule, as we detected no altered transcriptome, therefore, no altered expression of NF κ B- or p38/JNK-pathway target genes in BCMA-deficient ASCs.

It would also have been interesting to investigate NF κ B signalling in BCMA⁺ and KO ASCs, in presence or absence of APRIL and BAFF stimulation.

We have not investigated NF κ B signaling in BCMA⁺ and -deficient ASCs because we did not find any changes in genes involved in the NF κ B signaling pathway in transcriptome analysis in ASCs from BCMA-KO Δ 3 mice when compared to ASC from WT mice, which confirmed that APRIL signaling pathways are not changed in the absence of BCMA. This finding corroborates our hypothesis that the primary role of BCMA is not as a signal-transducing membrane receptor.

Minor points:

a) The authors state that DN cells are mostly IgG⁺ ASCs. The data to support this should be shown.

In a previous publication from our group (Pracht et al., 2017, EJI), we showed by intracellular staining that the frequency of IgM-/IgA-negative ASC is comparable to IgG-positive ASCs.

[REDACTED]

*Additionally, in our latest publication (Schulz et al., 2025, Frontiers in Immunol.), we aimed to perform a transcriptomic analysis of IgG ASCs. Repertoire reconstruction of sorted IgA-/IgM-negative ASCs confirmed the purity of the sorted plasma cells and revealed that >90% of clones produced transcripts encoding IgG constant regions. We have included both citations in the **legend of Fig. 1C**.*

b) Line 179: Decrease, not decreased.

We have changed "increased" to "increase" in line 179.

Comments to the Reviewers

Menzel et al.

Note: Reviewer comments are in BLACK, and authors' responses are in RED and ITALICS

Reviewer #1

The authors have adequately addressed most of my comments experimentally or by discussing their results. I have no further comments.

Reviewer #2

No additional comments of the revised manuscript.

Reviewer #3

The authors have provided additional data and have modified the manuscript in response to the reviewers comments. My view is that they have adequately replied to the queries raised. The methodology is sound and the results settle previous controversies about the function of BCMA, an important target for immunotherapy.

Authors Response:

We thank all three reviewers for their feedback and for the time and effort they invested in evaluating our manuscript. Their constructive comments have strengthened our manuscript and we appreciate their endorsement of the experimental approach and clarity of our findings.